| Open Peer Review | Bacteriology | Methods and Protocols

# A rapid method to simultaneously separate bacterial and eukaryotic RNA during infections reveals increased intracellular expression of *Staphylococcus aureus* and *Shigella flexneri* virulence factors

Duarte N. Guerreiro,[1,2,3,4] Johan Henriksson,[2,3,5,6] Jörgen Johansson[1,2,3]

**ABSTRACT** Transcriptome analysis has become an increasingly reliable method to assess the response of microorganisms to the environment inside the host cell. However, to maximize the reading depth of the pathogen and host transcriptomes, physical separation of the RNA pools is preferred, particularly to determine intracellular gene expression of the pathogen. Here, we set out to determine the intracellular gene expression of two important pathogens, the Gram-positive bacterium *Staphylococcus aureus* and the Gram-negative bacterium *Shigella flexneri*. For accurate determination, we developed a rapid method to physically separate bacterial from eukaryotic RNA with a high level of purity. Analysis by RT-qPCR demonstrated that bacterial and eukaryotic RNA could be separated efficiently, enriching the bacterial RNA pool >20-fold. Comparing gene expression of RNA extracted from different purification fractions by RNAseq showed an upregulation of different *S. aureus* genes. Among these was *vraX*, which encodes a secreted peptide that binds the C1q protein in the classical complement pathway.

**IMPORTANCE** Infectious diseases are one of the largest causes of deaths world-wide despite access to antimicrobials and vaccines. To develop new strategies to defeat microbial infections, a greater understanding of the infection process is needed such as analyzing the microbial and host responses during different stages of infection. Several microbes can invade host cells and being able to accurately monitor their and the host cells' gene expression is critical. Here, we have developed an easy and inexpensive method to reliably enrich for and separate bacterial and eukaryotic RNA after an intracellular infection. Our method is applicable to both Gram-negative and Gram-positive bacteria. Using this method, we have identified several genes to be upregulated during S. aureus infection of macrophage cells. Our data could prove useful to obtain new strategies for developing antimicrobial drugs.

**KEYWORDS** intracellular bacteria, *Staphylococcus aureus*, *Shigella flexneri*, macrophages, RNA separation, RT-qPCR, RNAseq, gene expression

B acterial pathogens and their eukaryotic hosts are required to respond accurately to the presence of each other during infection by reshaping their respective transcriptional landscapes. Thus, an in-depth examination of the host-pathogen transcriptomes is essential to determine how the organisms respond during an infection. With current sequencing technologies, transcriptome changes can be monitored during infection. However, the simultaneous analysis of host and pathogen transcriptomes is limited by the maximum number of reads available in the sequencing platforms

Address correspondence to Jörgen Johansson, jorgen.johansson@umu.se.

The authors declare no conflict of interest.

See the funding table on p. 13.

(1). This is particularly troublesome for RNA of intracellular pathogens since the level of such transcripts is typically much lower compared to the RNA of the eukaryotic host cell. To overcome this caveat, targeted degradation of eukaryotic RNA is possible through commercial kits. Yet, their efficacy may vary depending on the sample type (2). Differential lysis methods are also often employed although such methods typically lead to reduced total RNA recovery and loss of the RNA from one of the organisms (3–9). On the other hand, the Dual-Seq method offers a different approach by allowing a simultaneous analysis of the host and pathogen followed by an *in silico* allocation of the transcriptomes to the respective organisms (10, 11). The dominant quantities of eukaryotic RNA in the sample and the limiting number of reads by the sequencing platforms may ultimately compromise an in-depth analysis of the bacterial transcriptome (8). Thus, a different approach to efficiently separate bacterial from eukaryotic RNA is desirable to reliably assess the expression of particularly bacterial RNA during an infection. To better understand intracellular virulence gene expression in two bacterial pathogens, the Gram-positive bacterium *Staphylococcus aureus* and the Gram-negative bacterium *Shigella flexneri*, we developed a quick, efficient, and inexpensive method to physically separate bacterial and eukaryotic RNA during intracellular infection. We infected *Mus musculus* J744A.1 macrophages with either *S. aureus* or *S. flexneri*. Infected cells were lysed maintaining bacteria viable, different fractions separated, and the RNA of each fraction extracted (Fig. 1 and 2). We determined the purity of the extracted RNAs in each fraction by RT-qPCR and observed that more than 95% of the eukaryotic RNA was removed from the pelleted bacterial fractions, while the eukaryotic supernatant fraction eliminated 99.5% of the bacterial RNA (Fig. 3 and 4). Using the method, we compared *S. aureus* gene expression of bacteria grown in broth culture flasks with bacteria grown intracellularly by both RT-qPCR and RNAseq (Fig. 5). Our results showed a strong upregulation of several genes, both known *S. aureus* virulence-associated genes and also yet-to-be-characterized genes. We imagine that this method could assist our understanding of the intracellular gene expression of pathogens.

## MATERIALS AND METHODS

### Strains, growth, and infection conditions

*M. musculus* macrophages J774A.1 (ATCC) were incubated to full confluence in 6-well plates with growth medium (Advanced DMEM [Gibco] supplemented with GlutaMAX [Gibco] and 10% fetal bovine serum [Sigma]) at 37°C with 5% $CO_2$. For routine growth, *S. aureus* MSRA1369 strain (12) was grown in BHI (Becton) and *S. flexneri* M90T strain (13) was grown in TSB (Acumedia). Bacterial cultures were incubated for 16 h at 37°C in their respective mediums. Cultures of *S. aureus* were then diluted to 1:100 in fresh medium, while cultures of *S. flexneri* were diluted to 1:10 in fresh medium and incubated for 1 h to achieve the desired cell density. Bacterial cultures were centrifuged at 14,000 × *g* for 1 min and resuspended in pre-warmed infection medium (Advanced DMEM [Gibco] supplemented with GlutaMAX [Gibco]). Infections were carried out with an MOI of 100:1 for 30 min before wells were washed twice with pre-warmed Dulbecco's Phosphate Buffer Saline (DPBS, Gibco) and further incubated with growth medium supplemented with gentamicin (50 µg.mL$^{-1}$) for 30 min. Cells infected with bacteria but not further processed were denoted "Infected macrophage sample" and were used as references for the efficacy of fractionation in the different experiments.

### Bacterial viability assessment

*S. aureus* and *S. flexneri* grown for 1 h, as described above, were centrifuged and resuspended in either PBS or ice-cold PBS/Triton. To simulate the approximate time taken between the disruption of the macrophages and plating or RNA extraction, bacteria resuspensions were kept in ice for 5 min, centrifuged at 7,000 × *g* for 5 min at 4°C, serially diluted in PBS, and plated in LB agar. Plates were incubated at 37°C for 24 h before determining CFU. Two biological replicates were used.

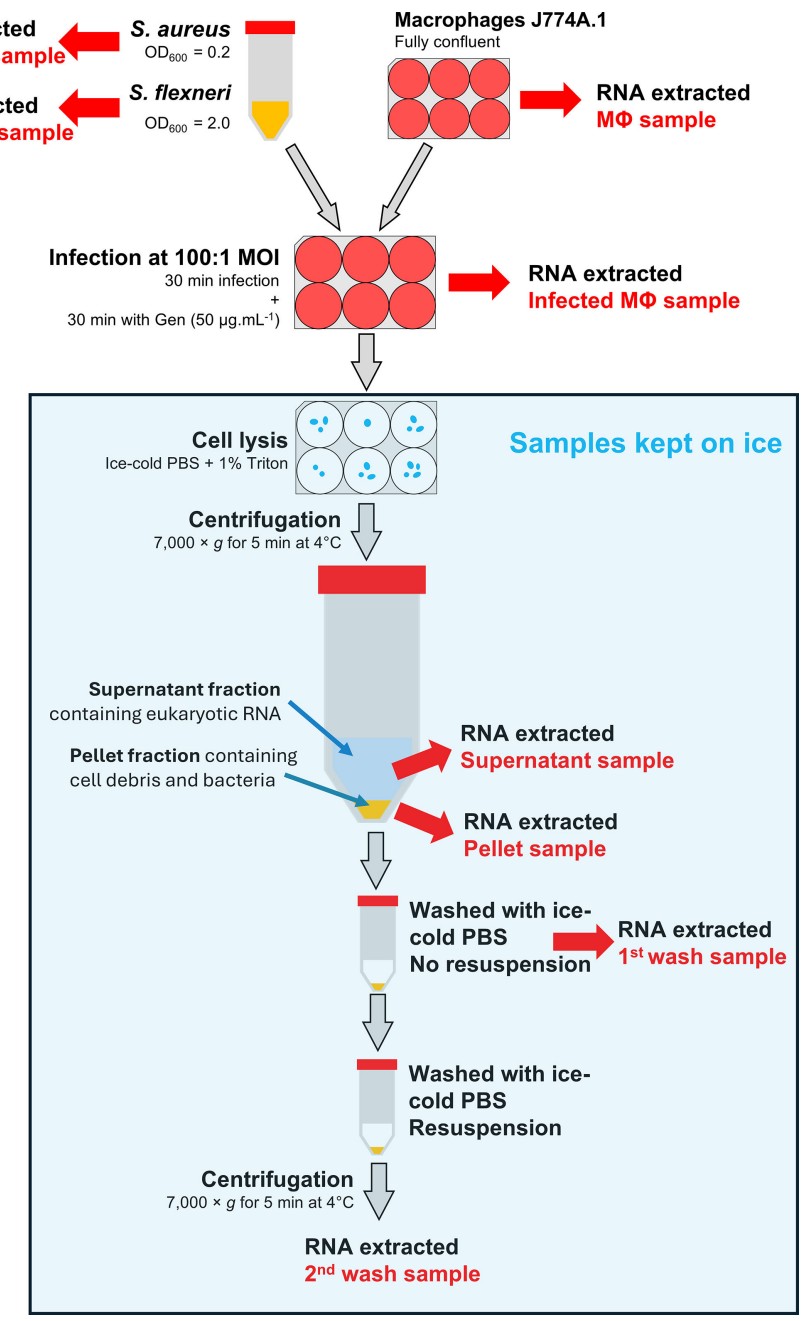

**FIG 1**  Schematic of method. Please see text for details.

## Physical separation of bacterial and eukaryotic RNAs

Infected macrophages were washed twice with DPBS before 1 mL of ice-cold PBS/Triton was added to each well. The solution was pipetted up and down several times to detach and lyse the eukaryotic cells. From this point, samples were kept on ice and centrifuged at 4°C to inhibit RNase activity as well as eukaryotic and bacterial gene expression. The cell lysate was centrifuged at 7,000 × $g$ for 5 min. The supernatant and pellet fractions were recovered and used for RNA extraction (supernatant and pellet fractions, respectively). The pellet was washed carefully with 1 mL of ice-cold PBS without centrifugation (Pellet, Wash × 1). The pellet was washed once more, pipetted several times to detach the pellet from the bottom of the tubes, and centrifuged (Pellet, Wash × 2). Pellets (Unwashed, Wash × 1, and Wash × 2) were resuspended in 400 µL disruption buffer (10%

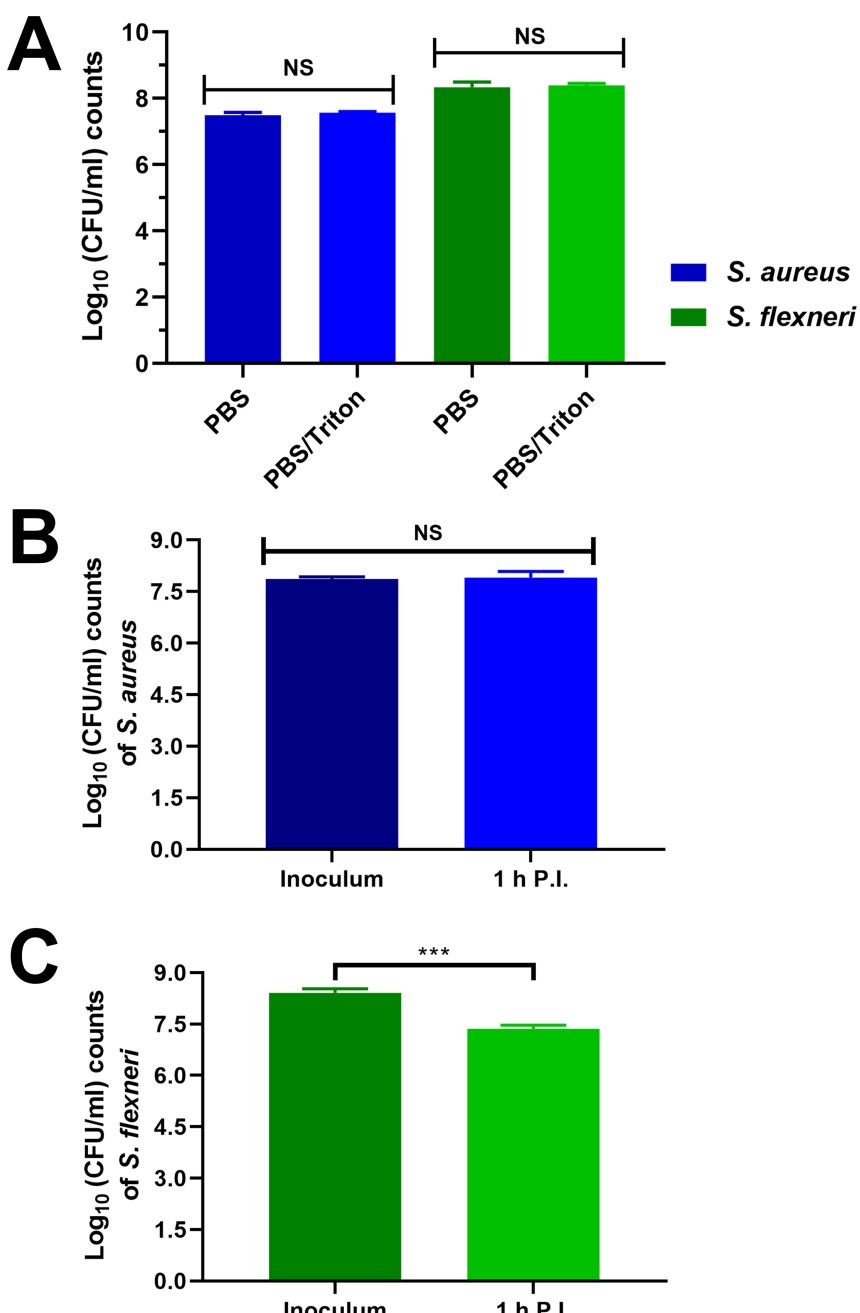

**FIG 2** Integrity of bacterial cells is not compromised by ice-cold PBS/Triton method and efficiently retrieves intracellular bacteria. (A) CFU counts of *S. aureus* or *S. flexneri* grown to late logarithmic growth phase before resuspension in PBS or ice-cold PBS/Triton followed by serial dilutions and plating on LB agar. CFU counts of the initial inoculum and internalized cells after 1 h of (B) *S. aureus* MRSA1369 and (C) *S. flexneri* M90T. Internalized bacteria were extracted with ice-cold PBS supplemented with 1% Triton X-100 before serial dilution and plating on LB agar. Statistical analysis was performed using paired Student's *t*-test (NS, non-significant; ***, *P* value of <0.001). Two biological replicates were used.

glucose, 12.5 mM Tris-HCl [pH 7.6], 5 mM EDTA), and their RNA was extracted (see below). For controls, 1 mL of bacterial cultures of *S. aureus* and *S. flexneri* was centrifuged at 14,000 × *g* for 30 s and resuspended in 400 µL of disruption buffer and used for RNA extraction. Controls for non-infected macrophages were made by adding 1 mL of

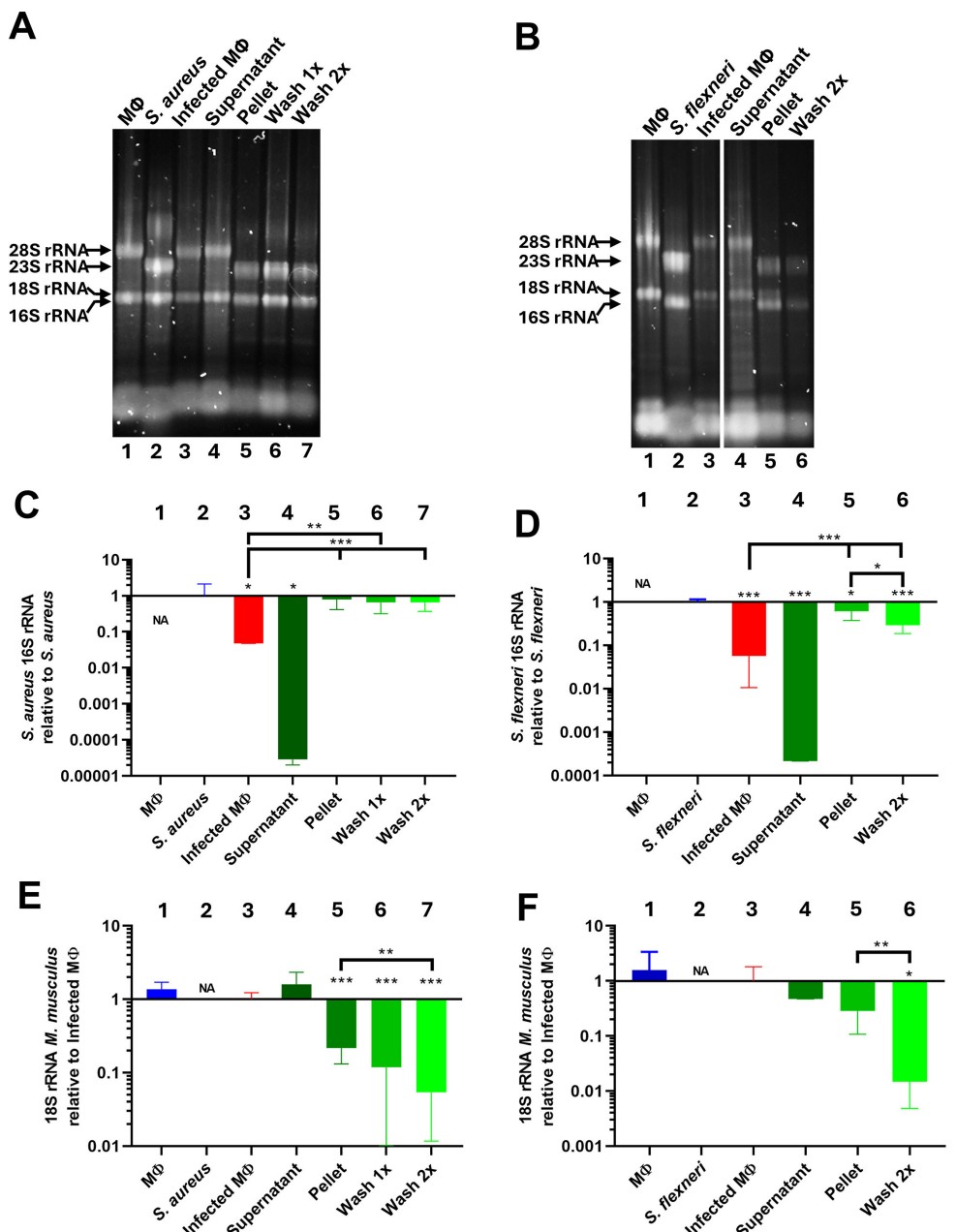

**FIG 3** Fractionating eukaryotic (*M. musculus*) and bacterial RNA enriches for bacterial RNA without compromising its function. (A and B) Separation and staining of total RNA obtained from indicated extraction steps. Levels of (C) *S. aureus* 16S rRNA, (D) *S. flexneri* 16S rRNA, or (E and F) levels of 18S rRNA from *M. musculus,* as determined by RT-qPCR. Samples shown in the different bars are bar 1, macrophage (MΦ); bar 2, *S. aureus* (A, C, and E) or *S. flexneri* (B, D, and F) grown in broth culture; bar 3, macrophages infected (infected MΦ) either with *S. aureus* (A, C, and E) or *S. flexneri* (B, D, and F), respectively; bar 4, supernatant fraction; bar 5, pellet fraction; bar 6, pellet fraction washed once (for *S. aureus* only) or pellet fraction washed twice. 16S rRNA levels from different fractions are shown relative to *S. aureus* or *S. flexneri* grown in broth culture, respectively (C and D, bar 2). 18S rRNA levels from different fractions are shown relative to macrophages infected with bacteria (E and F, bar 3). Statistical analysis was performed using paired Student's *t*-test with the infected macrophage sample as reference (*, *P* value of <0.05; **, *P* value of <0.01; ***, *P* value of <0.001). Three biological replicates were used.

disruption buffer to each well, cells were scraped, and 400 µL was used for RNA extraction. Three biological replicates were used.

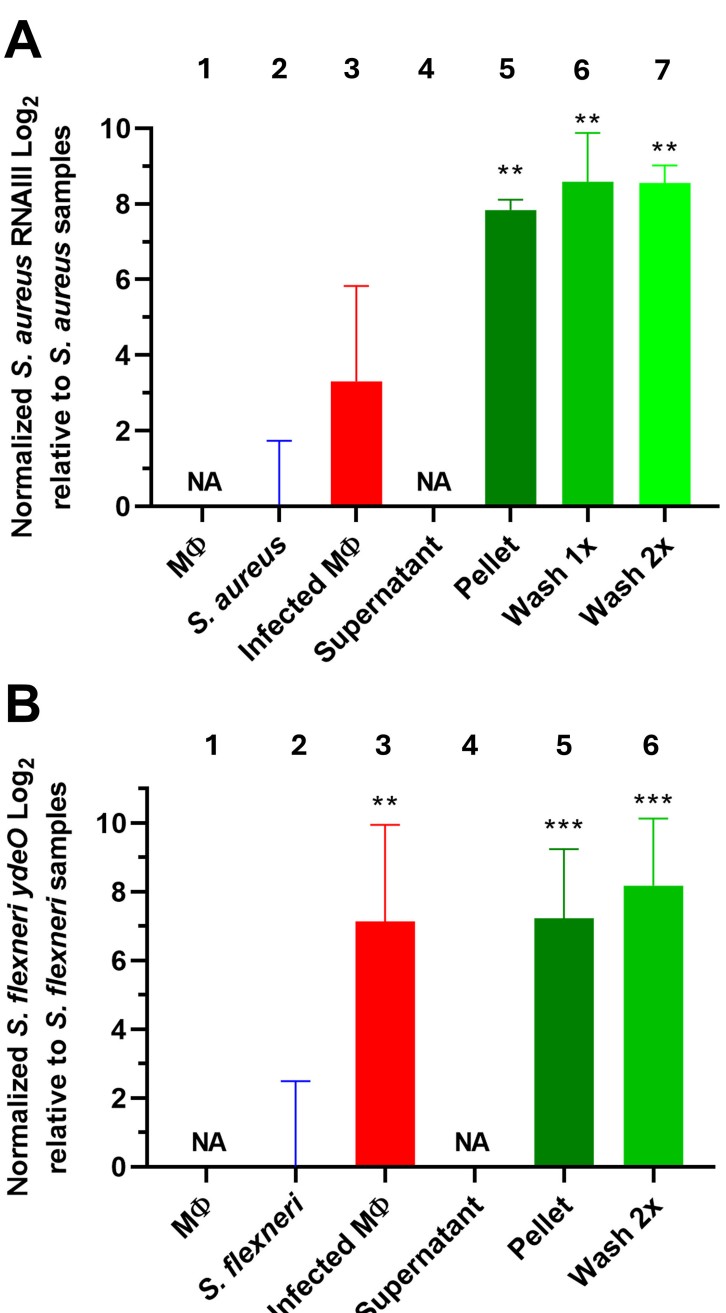

**FIG 4** Fractionating RNA samples of infected macrophages increases accuracy of virulence gene expression determination. Levels of (A) *S. aureus* RNAIII or (B) *S. flexneri ydeO* obtained from indicated extraction steps and determined using RT-qPCR. RNAIII and *ydeO* expression levels were determined using the 16S rRNA as reference gene before correlating the levels to *S. aureus* or *S. flexneri* from bacteria grown in broth culture flasks (A and B, bar 2, respectively). Samples in the different panels (A and B) were bar 1, macrophages (MΦ); bar 2, bacteria (*S. aureus*—A) or (*S. flexneri*—B) grown in broth culture flasks; bar 3, macrophages infected with either (A) *S. aureus* or (B) *S. flexneri* (infected MΦ); bar 4, supernatant fraction of infected macrophages after centrifugation; bar 5–7, pellet fraction and pellet fractions washed once (for *S. aureus* only) or twice of infected macrophages. Statistical analysis was performed using paired Student's *t*-test with the infected macrophage sample as reference (**, *P* value of <0.01; ***, *P* value of <0.001). Three biological replicates were used.

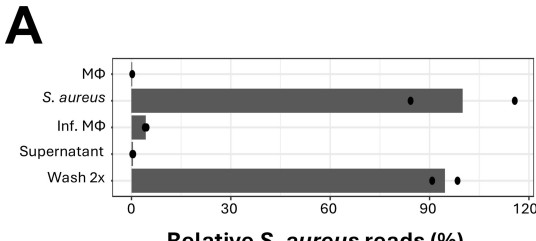

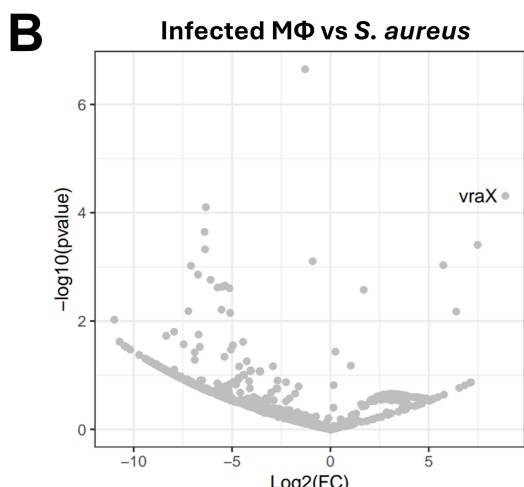

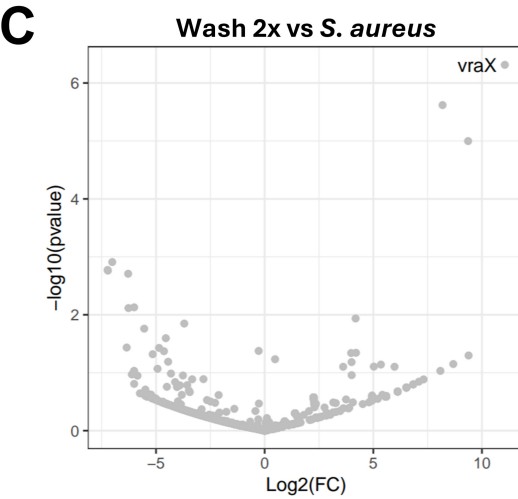

**FIG 5** Sequencing of *S. aureus* RNA from fractionated samples identifies *S. aureus* genes induced intracellularly. After infection of macrophages, samples were fractionated and RNA isolated and sequenced. (A) Ratios of *S. aureus* reads in different samples. 100 correspond to maximum amount (i.e., non-mapping reads are not counted). Error bars are ±s.d., where the variance is computed assuming a normal distribution in log-space. The variance of *S. aureus* grown in broth culture has been transferred to other samples, defining it as 100% ± 0. The amount of reads for each bar is an average of two independent biological samples, and each individual sample has been indicated by a dot. (B and C) Volcano plot, showing differentially expressed genes in Infected macrophages (infected MΦ) vs *S. aureus* grown in broth culture (*S. aureus*) as well as twice washed (Wash 2×) vs *S. aureus* grown in broth culture (*S. aureus*). Two biological replicates were used.

## RNA extraction

The total RNA was extracted from all samples by Trizol reaction as previously described (14). Briefly, bacterial pellets were resuspended in 400 µL of disruption buffer and transferred into a 2 mL screw-cap tube containing 500 µL of acid phenol (pH 4.5) (Invitrogen) and 500 µL of 0.1 mm zirconium beads (Biospec). Cells were disrupted at 6.0 m.s$^{-1}$ for 30 s in FastPrep-24 (MP Biomedicals). Samples were centrifuged at 12,000 × $g$ for 5 min at 4°C. The upper aqueous phase was transferred to a 2 mL tube containing 1 mL of TRI reagent solution (Thermo Fisher), mixed thoroughly, and incubated at room temperature for 5 min. Two hundred microliters of chloroform were added and vortexed. The mixtures were centrifuged at 12,000 × $g$ for 5 min at 4°C, and the upper phase was transferred to a new tube containing 500 µL of chloroform and vortexed. Samples were centrifuged once more, the upper phase was transferred to a new tube containing a 0.7× volume (of the sample) of isopropanol and mixed and incubated at −20°C for 1 h. Samples were washed with 500 µL of 80% ethanol, and the total RNA was resuspended in 50 µL of RNase-free water. Total RNA was resolved in 0.7% agarose gels, dyed with SYBR Gold (Thermo Fisher) for 15 min and pictures captured in Amersham Imager 680.

## RT-qPCR

Total RNA was quantified in Nanodrop (Thermo Fisher) and adjusted to 1 µg.µL$^{-1}$. Residual DNA was removed with ezDNase (Thermo Fisher), and cDNA was synthesized with SuperScript IV Reverse Transcriptase (Thermo Fisher) using random primers (Invitrogen) by following the manufacturer's recommendations. Gene expression was measured by RT-qPCR in CFX Connect Real-Time PCR Detection System (Bio-Rad) using QuantiTect SYBR Green PCR Kits (Qiagen). Primers sequence 5′-GGGGAATCAGGGT TCGATTCC-3′ and 5′-GGATCCTCGTTAAAGGATTTAAAGTGG-3′ were used for *M. musculus* 18S rRNA; 5′-GGCGGTTTTTTAAGTCTGATGTG-3′ and 5′-CGCACATCAGCGTCAGTTAC-3′ for *S. aureus* 16S rRNA, 5′-CTCTACTAGCAAATGTTACTCACTTGC-3′ and 5′-GAAGGAGTG ATTTCAATGGCACA-3′ for *S. aureus* RNAIII; and 5′-GGGGAATATTGCACAATGGGC-3′ and 5′-CTGCGGGTAACGTCAATGAG-3′ for *S. flexneri* 16S rRNA, 5′-CGGCAAAAATAGACAGGCAA GAA-3′ and 5′-GCGAAGTTTTTACGTGCTTCAAAT-3′ for *S. flexneri* ydeO, respectively. The annealing temperature used for all primers was 53°C. Three biological replicates were used.

## Transcriptome libraries and sequencing

A total of 10 µg of isolated RNA was treated with DNAse I (Roche) according to the manufacturer recommendations. RNA was cleaned with RNeasy (Qiagen) kit according to the manufacturer recommendations. RNA was fragmented with RNA Fragmentation Reagents (Thermo Fisher) at 70°C for 3.5 min. From this point, unless otherwise stated, all RNA samples were cleaned between each step with RNeasy (Qiagen) kit with the following modifications, and 100 µL of RNA was added to 350 µL of RLT buffer and 550 µL of 96% ethanol. Samples were dephosphorylated with Shrimp Alkaline Phosphatase (NEB), followed by the 5′ phosphorylation with T4 Polynucleotide Kinase (NEB) according to the manufacturer recommendations. RNA solution was concentrated to 4.5 µL, mixed with 1 µL of 15 mM PAGE purified 5′RA (5′-UCCCUACACACGACGCUCUU CCAUCU-3′), heated for 3 min at 65°C, and cooled on ice. The ligation reaction was carried out with T4 RNA Ligase (NEB) kit and 10% DMSO and incubated for 2 h at room temperature. Samples were separated by size on a 6% denaturing polyacrylamide gel, and RNA fragments ranging in size from 100 to 400 nucleotides were recovered from the gel and purified with Costar Spin-X (Merck) and ethanol precipitated. RNA fragments were ligated to Adenylated 3′ DA adaptor (5phos-AGATCGGAAGAGCACAGTCTGAACTC CAG-3ddC) using truncated T4 RNA Ligase 2 (NEB) supplemented with 25% PEG-8000 (final concentration) and incubated at room temperature for 2 h. Non-ligated 3′DA was removed using 1.6× vol:vol AMPure XP beads (Beckman Coulter). RNA was converted to cDNA by mixing samples with RT primer (5′-CTGGAGTTCAGACGTGTGCTCTTCCGATC

T-3′) and reverse transcription buffer, heating to 80°C for 2 min and cooling to room temperature for 5 min. cDNA was synthesized with SuperScript IV Reverse Transcriptase (Thermo Fisher) following the manufacturer recommendations. cDNA was purified by ethanol precipitation with the addition of 2 µL of GlycoBlue Coprecipitant (Thermo Fisher). cDNA was enriched using Phusion high-Fidelity PCR Master Mix (Thermo Fisher) by using the index primers containing similar sequences to the TruSeq adapters (RNA PCR Index primers 1 to 8 and 12, Illumina). Libraries were further enriched for six more cycles using enrichment primers (F—5′-AATGATACGGCGACCACCGAGATC-3′, R—5′-CAAGCAGAAGACGGCATACGAGAT-3′). PCR products were cleaned with QIAquick PCR purification kit (Qiagen). Libraries concentrations were determined with Qubit (Thermo Fisher) and adjusted to 15 pM by following Miseq system (Illumina) recommendations. Libraries were sequenced in the MiSeq sequencing platform (Illumina) in $2 \times 76$ bp paired-end mode. Two biological replicates were used.

### RNA-seq analysis volcano plots and pseudocounts

A common reference genome was generated by concatenating Mus_musculus.GRCm39.115 and *S. aureus* GCF_024172245.1_ASM2417224v1. The reads were trimmed using FASTP (v0.23.4) (15) and then aligned using STAR (v.2.7.11b, parameters --outMultimapperOrder Random) (16). The features were counted using htseq-count (v.2.0.7, parameters: -f bam -r pos -t gene -i gene_id) (17). Differential expression analysis was performed using DESeq2 v.1.44.0 (18).

   Fraction of *S.a.* read was computed as the count of aligned reads vs the total number of reads in the FASTP-trimmed file.

### Results calculations

gDNA of *M. musculus*, *S. aureus,* and *S. flexneri* was extracted with DNeasy Blood & Tissue Kit (Qiagen) and serially diluted in factors of 10 and used in RT-qPCR with the same pairs of primers as shown above. The obtained Cq values were used to calculate a linear regression for each target gene. The linear regressions were applied to the Cq values obtained from the cDNA of each gene to determine their expression. Results were expressed as a difference relative to infected macrophages. The expression of *S. aureus* RNAIII and *S. flexneri ydeO* was calculated using the Pfaffl relative expression formula (19) where 16S rRNA was used as an endogenous reference control for each sample before being correlated with the value from the *S. aureus* or the *S. flexneri* sample, respectively.

### Statistical analysis

Student's *t*-test was used for all statistical analysis on the GraphPad software.

### RESULTS

### The viability of bacteria is not compromised by the PBS/Triton lysis method

To extract bacterial cells from macrophages, ice-cold PBS/Triton was used to lyse the eukaryotic cells (Fig. 1). Since previous studies have shown that the bacterial cell wall can be differently affected by lysis methods, we first assessed whether the PBS/Triton lysis method affected the viability of different bacteria (6, 20). Furthermore, to massively decrease the bacterial transcription rate and mitigate any effect of the extraction protocol in the transcriptional landscape of the bacteria, we kept all solutions and samples at ice-cold temperatures and handled the samples swiftly (21). Treatment of Gram-positive *S. aureus* and Gram-negative *S. flexneri* with ice-cold PBS/Triton did not affect bacterial viability (Fig. 2A). To assess whether the lysis method altered the viability of bacteria infecting eukaryotic cells, CFU was determined in whole cell lysates during infection. Macrophages infected with either *S. aureus* or *S. flexneri* were lysed with PBS/Triton at 60 min post-infection (p.i.). Compared to their inoculum, approximately 90% of *S. aureus* and 10% of *S. flexneri* were isolated from macrophages 60 min p.i., being in line

with previous infection experiments (Fig. 2B and C [22, 23]). Overall, our results showed that ice-cold PBS/Triton could be used for cell lysis without affecting bacterial viability.

## Low-speed centrifugation and ice-cold PBS washings efficiently separate eukaryotic and bacterial RNA pools

Since bacteria were still intact after cell lysis, we sought to examine whether centrifugations and ice-cold PBS washings could be used to separate eukaryotic and bacterial RNA. An initial centrifugation was made, and the supernatant and pellet fractions were isolated. We hypothesized that the supernatant fractions would mainly contain eukaryotic cytoplasmic RNA, whereas the intact bacteria would reside in the pellet fraction (Fig. 1). To examine this, RNA was extracted from non-infected macrophages; bacteria grown in broth culture; infected macrophages; the supernatant fraction of infected macrophages; and the pellet fraction of infected macrophages, respectively, before visualization on agarose gels (Fig. 1, 3A and B). Since the amount of bacterial RNA in relation to eukaryotic RNA is low, bacterial 16S rRNA and 23S rRNA could not be visually detected in the infected and in the supernatant fractions, respectively (Fig. 3A and B, lanes 3 and 4). In contrast, the pellet fraction of macrophages infected with *S. aureus* contained almost exclusively 16S and 23S rRNA (Fig. 3A, lane 5), whereas the pellet fraction of macrophages infected with *S. flexneri* contained considerable amounts of eukaryotic 18S rRNA (Fig. 3B, lane 5). To examine whether the purity of the bacterial pellet fraction could be improved, washes of the pellet fractions were performed with ice-cold PBS. After two washes, no 18S rRNA could be detected in the pellet fractions (Fig. 3A, lanes 6 and 7 and Fig. 3B, lane 6). Our results suggest that ice-cold PBS and centrifugations could be used to efficiently separate pools of eukaryotic and bacterial RNA after infection.

## Low level of RNA cross-contamination between isolated fractions

Although the method presented here visually appears to separate eukaryotic and prokaryotic RNA, the RNA contamination between each fraction might be considerable. Also, the activity of the extracted RNA (i.e., the ability of the RNA to be used in downstream applications such as RT-qPCR or RNAseq) could not be assessed by visual inspection. Therefore, to further verify the RNA purity and activity in the different fractions, the highly abundant eukaryotic 18S rRNA and bacterial 16S rRNAs were targeted and quantified by RT-qPCR. We used an equal initial number of cells and bacteria for infection, as well as an identical amount of total RNA for the different RT-qPCR reactions, to directly be able to compare the levels of the rRNAs between fractions.

Comparing the amount of bacterial RNA isolated from bacteria grown in broth culture with bacteria that infected macrophages showed a 10- to 50-fold reduction of bacterial RNA in the latter sample, clearly highlighting that the pool of bacterial RNA is low inside infected cells (Fig. 3C and D, compare bars 2 and 3). The amount of isolated bacterial 16S rRNA was between 100- and 1,000-fold lower in the supernatant fractions when comparing that to the level of 16S rRNA in the infected macrophage samples (Fig. 3C and D, compare bars 3 and 4). On the other hand, the level of bacterial 16S rRNA (both for *S. aureus* and *S. flexneri*) increased ~25-fold in the unwashed pellet fraction as compared to the amount of 16S rRNA identified in the infected macrophage sample (Fig. 3C and D, compare bars 3 and 5). Further washing of the pellet did not markedly affect the levels of the *S. aureus* 16S rRNA but significantly decreased *S. flexneri* 16S rRNA (Fig. 3C and D). The decrease of *S. flexneri* RNA could affect downstream results interpretations, emphasizing the importance of always correlating expression of an RNA of interest with a solid endogenous control (e.g. 16S rRNA).

We next evaluated the amount of eukaryotic 18S rRNA in the different fractions. The amount of 18S rRNA in the supernatant fractions of macrophages infected with either *S. aureus* or *S. flexneri* was approximately as high as observed in the infected macrophage samples, which is not surprising considering that most eukaryotic RNA is cytosolic (Fig.

3E and F, compare bars 3 and 4). However, unwashed pellets of macrophages infected with *S. aureus* contained 20% of 18S rRNA compared with the infected macrophage sample, and single and double washes decreased this value further (Fig. 3E, compare bars 3 with bars 5, 6, and 7). When comparing with the infected macrophage sample, the amount of 18S rRNA in the unwashed pellet fraction of macrophages infected with *S. flexneri* was marginally reduced but could be further decreased in the pellet fraction washed twice (Fig. 3F, compare bar 3 with bars 5 and 6).

Conclusively, our results demonstrate that centrifugations and washes with ice-cold PBS efficiently separate RNA of different origins (eukaryotic or bacterial) and greatly reduce cross-contamination between fractions, are efficient for both Gram-positive and Gram-negative bacteria, and that the extracted RNA can be used in downstream applications such as RT-qPCR.

## Virulence-associated factors in bacteria are upregulated during an infection

During an infection, the expression of different bacterial and eukaryotic factors is altered. For *S. aureus* and *S. flexneri*, the virulence-associated genes, RNAIII and *ydeO*, respectively, are upregulated when the bacteria are internalized in phagocytic cells (24, 25). To verify and assess our protocol, the expression of RNAIII and *ydeO* was determined in the different fractions isolated from macrophages infected with *S. aureus* or *S. flexneri*, respectively. Since the number of bacterial reads vs eukaryotic reads greatly increased in the pelleted phases, we hypothesized that the accuracy of analyzing bacterial virulence gene expression would be more precise by using the pelleted phase as compared to the non-fractionated samples. Comparing *S. aureus* grown in broth culture flasks, with non-fractionated macrophages infected with *S. aureus* indicated that the level of RNAIII was ~10-fold upregulated inside cells (Fig. 4A, compare lanes 2 and 3). However, the error bar was large, and the results were not significant. Comparing the fractionated samples (unwashed and washed pellets) of macrophages infected with *S. aureus*, with *S. aureus* grown in broth culture flasks, an >100-fold upregulation of RNAIII expression was observed in the fractionated samples (Fig. 4A, compare lane 2 with lanes 5, 6, and 7). Similar results were observed when analyzing *ydeO* expression in *S. flexneri* infecting macrophages; *ydeO* expression was upregulated ~100-fold in both the infected non-fractionated macrophages and the pellet fractions of infected macrophages when compared with *S. flexneri* grown in broth culture flasks (Fig. 4B, compare lane 2 with lanes 5 and 6).

## Global analysis of RNA levels shows genes being up- or downregulated during infection

Since the RT-qPCR results agreed with previous observations, it prompted us to perform a global RNA sequencing analysis to analyze intracellular bacterial gene expression. For this purpose, we infected J774.1 cells with *S. aureus* as before and isolated RNA from the different fractions: macrophages only; bacteria grown in broth culture; infected macrophages; supernatant as well as pellet washed twice, respectively. The RNA expression pattern correlated between the infected macrophage sample and the washed sample as determined by alignment and counting reads per gene (Fig. S1). Also, the number of bacterial reads in the washed samples was considerably higher than the number of bacterial reads identified in the infected macrophage sample to a level almost equal to *S. aureus* grown in broth culture (Fig. 5A). This again shows that the method efficiently enriches bacterial RNAs.

When correlating RNA levels of the intracellular bacteria levels with RNA from bacteria grown in broth culture (Fig. 5B and C), several RNAs were up- or downregulated (Fig. 5B and C; Tables S1 and S2). The RNA showing the strongest induction in both the infected macrophage sample and the washed sample as compared with *S. aureus* grown in broth was *vraX*, encoding VraX, a secreted protein that has been shown to inhibit the classical complement pathway by binding C1q (26). The genes being downregulated intracellularly as compared with bacteria grown in broth were almost exclusively

encoding different housekeeping proteins, many of them ribosomal proteins (Tables S1 and S2).

## DISCUSSION

Following gene expression of bacteria and the eukaryotic cell during an infection is essential to understand the complex bacterial: cell interplay. However, the simultaneous analysis of both partners is difficult due to technical limitations. One such difficulty is the analysis of gene expression of either the host or the intracellular pathogen by RNAseq since the presence of the other organism might perturb the data and limit the analysis due to an unspecific or limited number of reads (1). This is particularly troublesome when analyzing the gene expression in the bacterial pathogen since the level of pathogen RNA often is much lower compared to the host RNA (8, 10).

To accurately assess intracellular gene expression of two different pathogens, *S. aureus* and *S. flexneri*, we developed a simple, but efficient method to physically separate bacterial and eukaryotic RNAs in large quantities with a high level of purity, while maintaining the functionality of the RNA for downstream applications. Importantly, the method can be applied to both Gram-positive and Gram-negative bacteria. Furthermore, the physical separation of host and pathogen RNA makes the removal of eukaryotic RNA that is required by different commercial kits redundant. The methods also remove degraded eukaryotic RNA, which usually cannot be targeted by hybridization probes (3). Importantly, we could show that the method preserves the functionality of the RNA and performed in-depth analysis, RT-qPCR, and RNA-sequencing to determine intracellular virulence gene expression. Based on our results, we also suggest that different more elaborate RNA-sequencing methods could benefit from a physical separation of RNA, such as methods to identify targets of small RNAs and techniques to structurally probe RNA (27). Other non-commercial protocols able to isolate RNA have been published. However, these protocols mainly focus on extracting RNA from Gram-positive bacteria (*S. aureus*) or from both Gram-positive and Gram-negative bacteria and not from the host, thus perturbing the possibility of following host and bacterial gene expression simultaneously (*S. aureus* and *S.* Typhimurium) (5, 6). The latter protocol also includes phenol inactivation at the first step, preventing isolation of host RNA. Through our method, intact and viable bacteria can be isolated. This could be an advantage when analyzing other aspects of bacterial infection, such as proteomics or lipidomics. For optimal results, it should be stressed that the presented protocol requires ice-cold temperatures at all stages as well as rapid handling of the samples.

Using the method, we compared the gene expression in bacteria grown in flasks with bacteria grown intracellularly and observed an upregulation of different transcripts. For instance, in both the unprocessed infected macrophage sample and in the washed sample, the gene *vraX* was the most highly induced when compared with RNA isolated from bacteria grown in broth culture (Fig. 5B and C; Table S1 and S2 [26]). VraX encodes a secreted protein of 55 amino acids that has been shown to inhibit the classical complement pathway by binding C1q. A strain lacking VraX shows a reduced ability to infect mice and also to block the complement-associated cell lysis of red blood cells. Our results suggest that once *S. aureus* enters macrophages, it increases the expression of VraX to be able to face the subsequent action of the complement system. Interestingly, VraX is highly upregulated when the cell wall of *S. aureus* is affected (28). For instance, *vraX* was upregulated >200-fold when vancomycin-sensitive *S. aureus* was exposed to vancomycin (29). This implies that the cell wall of *S. aureus* is affected when the bacterium becomes intracellular although the exact mechanism by how the expression of *vraX* is induced remains to be identified.

Although most genes show a similar expression pattern both in the unprocessed infected macrophage sample and in the washed sample, we also expect differences: Bacterial RNA isolated from unprocessed infected macrophages will have relatively few reads in relation to eukaryotic RNA, thus increasing the sample variation when compared with the washed samples (Fig. 4; Tables S1 and S2).

A shortcoming of this study is that it has only been performed in cultured cells and not in more complex cell structures such as organoids, whole organs, or animals, which contain several different types of cells. It will, thus, be of interest to examine if the *S. aureus* expression pattern will be similar also in other cell types, such as epithelial cells and if the method could be applied to separate bacterial and eukaryotic RNA from *in vivo* tissue samples. We believe that our method may be applied to examine gene expression in different bacteria during different stages of infection. One important aspect that could affect the reliability of the data is how long the bacterium resides within the vacuole of the host cell. Some bacteria (e.g., *Salmonella enterica* serovar Typhimurium) can reside within vacuoles (Salmonella containing vacuoles—SCV) for a long time, whereas other bacteria (e.g., *Listeria monocytogenes*) escape the vacuole relatively quickly (30, 31). It is, therefore, advisable to adjust centrifuge speed to allow maximal bacterial RNA recovery.

Conclusively, we present a reliable and rapid method to separate bacterial and eukaryotic RNAs during an early *in vitro* infection scenario, which we herein show can be used to enhance in-depth analysis of host-pathogen interactions.

## ACKNOWLEDGMENTS

We thank Dr. A. Puhar for supporting us with *S. flexneri* strain M90T. We thank Dr. M. Milovojevic for critical reading of the manuscript.

J.J. was funded by the Swedish Research Council grants #2020-02005_3 and #2023-02679, Umeå University, the Stiftelsen Olle Engkvist Byggmästare, Vinnova grant 2019-05491, and the Erling-Persson Foundation. J.H. is supported by the Swedish Research Council grants #2021-06602, #2024-03952, the Swedish Cancer Society #23 3102 Pj, and the Swedish Foundation for Strategic Research #ITM24-0035.

## AUTHOR AFFILIATIONS

[1]Laboratory for Molecular Infection Medicine Sweden, Umeå University, Umeå, Sweden
[2]Department of Molecular Biology, Umeå University, Umeå, Sweden
[3]Umeå Centre for Microbial Research, Umeå University, Umeå, Sweden
[4]Centre for Ecology, Evolution and Environmental Changes (cE3c) & Global Change and Sustainability Institute (CHANGE), Faculdade de Ciências, Universidade de Lisboa, Lisboa, Portugal
[5]SciLifeLab, Umeå university, Umeå, Sweden
[6]IceLab, Umeå university, Umeå, Sweden

## PRESENT ADDRESS

Duarte N. Guerreiro, Department of Biology, Faculdade de Ciências, Universidade de Lisboa, Lisboa, Portugal

## AUTHOR ORCIDs

Duarte N. Guerreiro http://orcid.org/0000-0002-3953-1752
Johan Henriksson http://orcid.org/0000-0002-7745-2844
Jörgen Johansson http://orcid.org/0000-0002-0904-497X

## FUNDING

| Funder | Grant(s) | Author(s) |
|---|---|---|
| Vetenskapsrådet | #2020-02005_3, #2023-02679, #2021-06602, #2021-06602 | Jörgen Johansson |
| Umeå Universitet | | Jörgen Johansson |
| Olle Engkvists Stiftelse | | Jörgen Johansson |
| VINNOVA | 2019-05491 | Jörgen Johansson |

| Funder | Grant(s) | Author(s) |
|---|---|---|
| Erling-Perssons Stiftelse | | Jörgen Johansson |
| Cancerfonden | #23 3102 Pi | Johan Henriksson |
| Stiftelsen för Strategisk Forskning | #TM24-0035 | Johan Henriksson |

## AUTHOR CONTRIBUTIONS

Duarte N. Guerreiro, Conceptualization, Data curation, Investigation, Methodology, Software, Validation, Visualization, Writing – review and editing | Johan Henriksson, Data curation, Formal analysis, Investigation, Methodology, Software, Validation, Visualization, Writing – review and editing | Jörgen Johansson, Conceptualization, Funding acquisition, Project administration, Resources, Writing – original draft, Writing – review and editing

## DATA AVAILABILITY

All the R code is available at Github (https://github.com/henriksson-lab/guer-reiroa_dualseq). Raw FASTQ data are available at Zenodo (https://doi.org/10.5281/zenodo.17632721).

## ADDITIONAL FILES

The following material is available online.

### Supplemental Material

**Fig. S1 (Spectrum03745-25-s0001.pdf).** Scatter plot.
**Table S1 (Spectrum03745-25-s0002.xls).** *S. aureus* in infected macrophages versus *S. aureus* grown in broth culture.
**Table S2 (Spectrum03745-25-s0003.xls).** Twice washed fraction versus *S. aureus* grown in broth culture.

### Open Peer Review

**PEER REVIEW HISTORY (review-history.pdf).** An accounting of the reviewer comments and feedback.

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
