## [Reviewer comments · Microbiology Spectrum]

Microbiology Spectrum

A rapid method to simultaneously separate bacterial and eukaryotic RNA during infections reveals increased intracellular expression of *Staphylococcus aureus* and *Shigella flexneri* virulence factors

Duarte Guerreiro, Johan Henriksson, and Jorgen Johansson

Corresponding Author(s): Jorgen Johansson, Umea Universitet

Review Timeline:

Submission Date:	November 20, 2025
Editorial Decision:	December 23, 2025
Revision Received:	January 5, 2026
Editorial Decision:	February 1, 2026
Revision Received:	February 3, 2026
Accepted:	February 27, 2026

Editor: Monica Cartelle Gestal

Reviewer(s): The reviewers have opted to remain anonymous.

Transaction Report:

DOI: <https://doi.org/10.1128/spectrum.03745-25>

Re: Spectrum03745-25 (**A rapid method to simultaneously separate bacterial and eukaryotic RNA during infections reveals increased intracellular expression of *Staphylococcus aureus* and *Shigella flexneri* virulence factors**)

Dear Prof. Jorgen Johansson:

Thank you for the privilege of reviewing your work. Below you will find my comments, instructions from the Spectrum editorial office, and the reviewer comments.

Dear Dr. Johansson,

Please revise the protocol and pay especial attention to the comments provided by reviewer #1. Given this manuscript has been submitted as protocol/method details are needed and clarity of language is critical.

Revision Guidelines

Sincerely,
Monica Cartelle Gestal
Editor
Microbiology Spectrum

Reviewer #1 (Comments for the Author):

The manuscript by Guerreiro et al, describes a fast and inexpensive method to separate bacterial and eukaryotic RNA for simultaneous transcriptomics. The method described is simple, relatively fast, and broadly applicable to both Gram(+) and Gram(-) bacteria. The authors use of RNA-seq proof-of-principle, in this case *vraX* gene, is compelling and biologically meaningful. However, some claims are overstated ("our results demonstrate that centrifugations and washes with ice-cold PBS efficiently separate RNA of different origins (eukaryotic or bacterial) without contamination between fractions") and should be revised. For Student's t-test statistical analyses, it is not clear how many biological replicates were used. Another major question is whether this method can be applied to intracellular bacteria growing in vacuoles for longer than 60 minutes. Therefore, the final conclusion ("we present a reliable and rapid method to separate bacterial and eukaryotic RNAs during an in vitro infection scenario") should be restated and different infection conditions for other intracellular bacteria should be acknowledged. There are also frequent small grammatical errors that need to be fixed: "visually appear" should be "visually appears"; "This again show" should be "This again shows"; "was fragment" should be "was fragmented", etc.

Reviewer #2 (Comments for the Author):

This paper describes a method for separating bacteria from eukaryotic RNA to facilitate transcriptomic analysis of intracellular bacteria. It tests the method in part using *S. aureus* and *S. flexneri*, before using the method to do full transcriptomics with *S. aureus* only. Gene expression analysis on the Eukaryotic host macrophages is not performed. The overall conclusions of this paper are supported by the data presented.

Comments:

Introduction lines 72-75 should point to the data that supports this statement.

We appreciate the input made by both reviewers to our work. We took in consideration the comments and suggestions and made several modifications to the manuscript.

Reviewer #1 (Comments for the Author):

The manuscript by Guerreiro et al, describes a fast and inexpensive method to separate bacterial and eukaryotic RNA for simultaneous transcriptomics. The method described is simple, relatively fast, and broadly applicable to both Gram(+) and Gram(-) bacteria. The authors use of RNA-seq proof-of-principle, in this case *vraX* gene, is compelling and biologically meaningful. However, some claims are overstated ("our results demonstrate that centrifugations and washes with ice-cold PBS efficiently separate RNA of different origins (eukaryotic or bacterial) without contamination between fractions") and should be revised.

We appreciate the comments made by the reviewer and have made several modifications throughout the text to incorporate the suggestions.

For Student's t-test statistical analyses, it is not clear how many biological replicates were used.

We added the number of replicates made for each experiment in the Material and methods and in the Figure legends sections.

Another major question is whether this method can be applied to intracellular bacteria growing in vacuoles for longer than 60 minutes. Therefore, the final conclusion ("we present a reliable and rapid method to separate bacterial and eukaryotic RNAs during an *in vitro* infection scenario") should be restated and different infection conditions for other intracellular bacteria should be acknowledged.

We appreciate the reviewer's question and have modified the manuscript accordingly. We suggest that our RNA separation method could be applied to different bacteria at different stages of infection. In the discussion section, we bring up how differences in time spent in the vacuole between two bacterial pathogens (i.e. *Salmonella enterica* serovar Typhimurium and *Listeria monocytogenes*) could affect the results and how this possibly could be addressed by altering centrifugation speed.

There are also frequent small grammatical errors that need to be fixed: "visually appear" should be "visually appears"; "This again show" should be "This again shows"; "was fragment" should be "was fragmented", etc.

We thank the reviewer for highlighting this. Several modifications were made throughout the manuscript to fix the grammatical errors.

Reviewer #2 (Comments for the Author):

This paper describes a method for separating bacteria from eukaryotic RNA to facilitate transcriptomic analysis of intracellular bacteria. It tests the method in part using *S. aureus* and *S. flexneri*, before using the method to do full transcriptomics with *S. aureus* only. Gene expression analysis on the Eukaryotic host macrophages is not performed. The overall conclusions of this paper are supported by the data presented.

Comments:

Introduction lines 72-75 should point to the data that supports this statement.

We appreciate the reviewer's comment and point out the figures supporting the statement.

Re: Spectrum03745-25R1 (**A rapid method to simultaneously separate bacterial and eukaryotic RNA during infections reveals increased intracellular expression of *Staphylococcus aureus* and *Shigella flexneri* virulence factors**)

Dear Prof. Jorgen Johansson:

Thank you for the privilege of reviewing your work. Below you will find my comments, instructions from the Spectrum editorial office, and the reviewer comments.

We would like to thank the authors for the modifications applied to the manuscript and while it has considerably improved, reviewer 2 has suggested a couple of more things to do. Please send us the revise version as early as your convenience.

Revision Guidelines

Sincerely,
Monica Cartelle Gestal
Editor
Microbiology Spectrum

Reviewer #2 (Comments for the Author):

This paper describes a method for separating bacteria from eukaryotic RNA to facilitate transcriptomic analysis of intracellular bacteria. It tests the method, in part, using *S. aureus* and *S. flexneri*, before using the method to do full transcriptomics with *S.*

aureus only. Gene expression analysis on the eukaryotic host macrophages is not performed. The overall conclusions of this paper are supported by the data presented.

Comments:

Figures 3a and 3b are unclear as presented. The labels of both these figures are identical and should be corrected. Additionally, the differences between lanes 3 and 4 mentioned in the text are not obvious. This would benefit from a clearer gel or additional labeling.

Figure 3D shows a significant loss of *S. flexneri* 16s rRNA after the second wash. Does this affect downstream results? This should be addressed in the text.

Figure 4 should be normalized to a stable endogenous control. The data in Figure 3D shows a reduction in 16srRNA between the pellet and 2x wash for *S. flexneri*, yet Figure 4B shows stable *ydeO*. This would imply that the measured relative *ydeO* expression may actually be increased after washing. Normalization would control for this.

Additional comment:

Figure 2 demonstrates that the 1% Triton treatment does not impact bacterial viability. The data would be even more convincing with rt-QPCR or RNA-Seq showing whether the 1% triton treatment followed by the pelleting and washing steps impacts gene expression.

Review part 1)

This paper describes a method for separating bacteria from eukaryotic RNA to facilitate transcriptomic analysis of intracellular bacteria. It tests the method, in part, using *S. aureus* and *S. flexneri*, before using the method to do full transcriptomics with *S. aureus* only. Gene expression analysis on the eukaryotic host macrophages is not performed. The overall conclusions of this paper are supported by the data presented.

Comments:

Introduction lines 72-75 should reference the data that supports this statement.

Figures 3a and 3b are unclear as presented. The labels of both these figures are identical and should be corrected. Additionally, the differences between lanes 3 and 4 mentioned in the text are not obvious. This would benefit from a clearer gel or additional labeling.

Figure 3D shows a significant loss of *S. flexneri* 16s rRNA after the second wash. Does this affect downstream results? This should be addressed in the text.

Figure 4 should be normalized to a stable endogenous control. The data in Figure 3D shows a reduction in 16srRNA between the pellet and 2x wash for *S. flexneri*, yet Figure 4B shows stable *ydeO*. This would imply that the measured relative *ydeO* expression may actually be increased after washing. Normalization would control for this.

Additional comment:

Figure 2 demonstrates that the 1% Triton treatment does not impact bacterial viability. The data would be even more convincing with rt-QPCR or RNA-Seq showing whether the 1% triton treatment followed by the pelleting and washing steps impacts gene expression.

Comments to editor)

The data largely supports the conclusion that this method can separate bacterial and eukaryotic RNA and even demonstrates that this could be useful for improving the depth and quality of bacterial gene expression analysis. However, additional controls/experiments would likely be needed to support any further conclusions about the specific gene expression changes in the paper.

The paper is written in standard english, but could benefit from further proofreading for minor typos/grammar issues.

Reviewer #2?

We are a bit surprised to see this comment:

“Introduction lines 72-75 should reference the data that supports this statement”

The comment was raised after the first submission by reviewer 2 and we answered the comment in the resubmission. Anyhow, we are indeed referencing the data in the manuscript (lines 71-78) that supports the statements we make.

Reviewer #2 (Comments for the Author):

This paper describes a method for separating bacteria from eukaryotic RNA to facilitate transcriptomic analysis of intracellular bacteria. It tests the method, in part, using *S. aureus* and *S. flexneri*, before using the method to do full transcriptomics with *S. aureus* only. Gene expression analysis on the eukaryotic host macrophages is not performed. The overall conclusions of this paper are supported by the data presented

We appreciate that the reviewer consider that the conclusions in the paper are supported by the data presented. We are also thankful for all comments made by the reviewer and have addressed them below.

Comments:

Figures 3a and 3b are unclear as presented. The labels of both these figures are identical and should be corrected. Additionally, the differences between lanes 3 and 4 mentioned in the text are not obvious. This would benefit from a clearer gel or additional labelling.

We appreciate the comments by the reviewer regarding the incorrect figure and also agree that the differences between the lanes are not obvious. We have now corrected the figure and rephrased the text (lines 251-254 in the “clean manuscript file”) to better support this notion.

Figure 3D shows a significant loss of *S. flexneri* 16s rRNA after the second wash. Does this affect downstream results? This should be addressed in the text.

We thank the reviewer for bringing this point up. Although the amount of 16S rRNA decreases, we consider the reduction to be of less importance for the interpretation of the results (i.e. fractionation of samples considerably increases the number of bacterial reads). Also, the reduction in 16S rRNA (70.9%), is less compared with the reduction in 18S rRNA from the bacterial fraction (more than 98.5%). Hence, since the total number of reads in downstream applications (e.g.: RNAseq) is limited, the even larger decrease of contaminating eukaryotic RNA in the Wash 2x fraction is advantageous for the quality of the results.

We however agree that this was not clearly stated in the text that the reduction in the 16S rRNA would affect downstream results and we have now addressed this issue in the

amended version of the manuscript, (lines 282-286 in the “clean manuscript file”). Also see the below point.

Figure 4 should be normalized to a stable endogenous control. The data in Figure 3D shows a reduction in 16srRNA between the pellet and 2x wash for *S. flexneri*, yet Figure 4B shows stable *ydeO*. This would imply that the measured relative *ydeO* expression may actually be increased after washing. Normalization would control for this.

We appreciate the reviewer’s comments and apologize for the unclarity in the text; this was not sufficiently described in the original submission. The data was indeed normalized to a stable endogenous RNA control (16S rRNA) in each sample before being correlated with the *S. flexneri* (and *S. aureus*) sample. The normalization step has now been better described (lines 210-213 and lines 458-461 in the “clean manuscript file”). This (together with the previous point) indicates that there is indeed a slightly (although not significantly) increased level of *ydeO* mRNA in the Wash 2 sample as compared with the Pellet sample.

Additional comment:

Figure 2 demonstrates that the 1% Triton treatment does not impact bacterial viability. The data would be even more convincing with rt-QPCR or RNA-Seq showing whether the 1% triton treatment followed by the pelleting and washing steps impacts gene expression.

We appreciate the reviewers’ comment, However, based on the experimental setup using ice-cold conditions, we consider ice-cold 1% Triton to have no or a very minute effect on the gene-expression pattern in bacteria. Even in *L. monocytogenes*, a bacterium able to grow at subzero temperatures, it takes days for the bacteria to start growing and activate the stress-sigma factor SigB at 4°C, Utratna *et al.*, 2014, doi.org/10.1155/2014/641647. This strongly suggest that at ice-cold conditions presented in this protocol for *S. aureus* and *S. flexneri*, the metabolic activity (and hence gene-expression) in these bacteria is close to zero. Hence, the sudden drop and sustained low temperatures would massively delay the bacteria’s ability to express the appropriate genes in response to external stress (including Triton exposure). In addition to this, bacteria were lysed quickly after the addition of ice-cold PBS/1% Triton, further narrowing the ability of the bacteria to alter their gene expression profile in response to Triton exposure.

We do however think that these points (the importance of timing and using ice-cold buffers) were not sufficiently described in the manuscript and the text has now been amended to better emphasize this (lines 229-232 and lines 376-378 in the “clean manuscript file”). The reference above has also been introduced in the manuscript.

Re: Spectrum03745-25R2 (**A rapid method to simultaneously separate bacterial and eukaryotic RNA during infections reveals increased intracellular expression of *Staphylococcus aureus* and *Shigella flexneri* virulence factors**)

Dear Prof. Jorgen Johansson:

Thank you for your patience and it is my pleasure to congratulate you for the acceptance of your manuscript. Please see the comments below

Your manuscript has been accepted, and I am forwarding it to the ASM production staff for publication. Your paper will first be checked to make sure all elements meet the technical requirements. ASM staff will contact you if anything needs to be revised before copyediting and production can begin. Otherwise, you will be notified when your proofs are ready to be viewed.

Sincerely,
Monica Cartelle Gestal
Editor
Microbiology Spectrum